# A breakthrough series collaborative to increase patient participation with hemodialysis tasks: A stepped wedge cluster randomised controlled trial

**James Fotheringham**[1], **Tania Barnes**[1], **Louese Dunn**[1], **Sonia Lee**[1], **Steven Ariss**[2], **Tracey Young**[2], **Stephen J. Walters**[2], **Paul Laboi**[3], **Andy Henwood**[3], **Rachel Gair**[4], **Martin Wilkie**[1]*

1 Sheffield Kidney Institute, Sheffield Teaching Hospitals NHS Foundation Trust, Sheffield, England,
2 School of Health and Related Research, University of Sheffield, Sheffield, England, 3 Renal Department, York Teaching Hospital NHS Foundation Trust, York, England, 4 Think Kidneys, UK Renal Registry, Bristol, England

* martin.wilkie@nhs.net

## Abstract

### Background

Compared to in-centre, home hemodialysis is associated with superior outcomes. The impact on patient experience and clinical outcomes of consistently providing the choice and training to undertake hemodialysis-related treatment tasks in the in-centre setting is unknown.

### Methods

A stepped-wedge cluster randomised trial in 12 UK renal centres recruited prevalent in-centre hemodialysis patients with sites randomised into early and late participation in a 12-month breakthrough series collaborative that included data collection, learning events, Plan-Study-Do-Act cycles, and teleconferences repeated every 6 weeks, underpinned by a faculty, co-production, materials and a nursing course. The primary outcome was the proportion of patients undertaking five or more hemodialysis-related tasks or home hemodialysis. Secondary outcomes included independent hemodialysis, quality of life, symptoms, patient activation and hospitalisation. ISRCTN Registration Number 93999549.

### Results

586 hemodialysis patients were recruited. The proportion performing 5 or more tasks or home hemodialysis increased from 45.6% to 52.3% (205 to 244/449, difference 6.2%, 95% CI 1.4 to 11%), however after analysis by step the adjusted odds ratio for the intervention was 1.63 (95% CI 0.94 to 2.81, P = 0.08). 28.3% of patients doing less than 5 tasks at baseline performed 5 or more at the end of the study (69/244, 95% CI 22.2–34.3%, adjusted odds ratio 3.71, 95% CI 1.66–8.31). Independent or home hemodialysis increased from

**Data Availability Statement:** A minimal dataset required to reach the conclusions drawn from this manuscript required the linkage of identifiable

patient information collected during the trial to Hospital Episode Statistics data, which at the time of writing is provided by the NHS Digital Data Access Request Service (NHS DARS, https://digital.nhs.uk/services/data-access-request-service-dars), and then appropriate processing. An application to NHS DARS can be submitted detailing lawful processing of the combined dataset and the period which HES data is required for. NHS DARS would verify appropriate permissions were in place as a result of this process. A data sharing agreement between the relevant parties would allow data to be transferred from the University of Sheffield to NHS DARS and on to those wishing to perform the enclosed analyses. Please contact ctru@sheffield.ac.uk for further information about the unlinked dataset which has the personal information required for linkage.

**Funding:** The Health Foundation Scaling Up Award.

**Competing interests:** JF has received speaker honoraria from Fresenius medical care and Novartis, and conducts research funded by the National Institute of Health Research (NIHR), Vifor Pharma and Novartis. MEW has received speaker honoraria Fresenius and Baxter, has acted on an advisory board for Baxter and has conducted research funded by the NIHR. SJW has received book royalties from Wiley and has received funds from NIHR, the Department of Health and Medical Research Council. This does not alter our adherence to PLOS ONE policies on sharing data and materials.

7.5% to 11.6% (32 to 49/423, difference 4.0%, 95% CI 1.0–7.0), but the remaining secondary endpoints were unaffected.

## Conclusions

Our intervention did not increase dialysis related tasks being performed by a prevalent population of centre based patients, but there was an increase in home hemodialysis as well as an increase in tasks among patients who were doing fewer than 5 at baseline. Further studies are required that examine interventions to engage people who dialyse at centres in their own care.

## Introduction

When it was first developed, hemodialysis (HD) was predominantly undertaken at home, however its widespread adoption as a treatment for end stage kidney disease (ESKD) in conjunction with the increasing age and multi-morbidity of the patients receiving it, has led to in-centre HD accounting for greater than 80% of dialysis in 79% of countries in the 2016 USRDS annual report [1]. Home HD (HHD) is associated with better survival, quality of life and lower costs compared with in-centre HD, and some of this advantage may relate to the enhanced knowledge, skills and confidence that these individuals have to self-manage their condition [2]. In recognition of the benefits of home dialysis The National Institute of Health and Care Excellence recommended of a prevalence of 10–15% HHD in the UK, and a US presidential executive order prioritised home dialysis therapies with financial incentives for providers to meet targets [3].

Consistently offering in-centre HD patients the opportunity to learn about and participate in their treatment potentially enables access to some of the health benefits that are associated with HHD as well as impacting on patient activation and health literacy while aligning with the goals of person-centred care [4,5]. Self-management programmes for in-centre hemodialysis patients have been associated with improvements in empowerment, perceived self-efficacy, medication adherence, phosphate control and interdialytic weight gain between dialysis sessions which all correlate with mortality and symptom burden [6–10].

Shared Hemodialysis Care (SHC) is an educational quality improvement initiative consistently provides the choice and opportunity for in-centre HD patients to learn about and engage in their own treatment. The HD process is broken down into approximately 14 component tasks, and patients are supported to participate at complexities and rates according to individual preference (Table 1).

**Table 1. Primary endpoint–five or more dialysis related tasks.**

| Patient preparation | Machine Preparation & Dialysis Initiation | During and after dialysis |
|---|---|---|
| Measuring weight<br>Measuring blood pressure and pulse<br>Measuring temperature<br>Washing hands<br>Preparing dressing (vascular access) pack | Lining dialysis machine*<br>Priming dialysis machine*<br>Programming dialysis machine<br>Needling fistula/graft or preparing tunnelled line*<br>Connecting lines to fistula/graft/tunnelled line and commencing dialysis* | Responding to machine alarms *<br>Disconnecting lines and completing dialysis*<br>Applying pressure to needle sites or locking tunnelled line<br>Giving your own anaemia injections (such as epoetin) |

* required tasks for independent haemodialysis endpoint.

In 2016 we initiated a program to scale up and spread SHC to 12 hospital sites in England. To do this we developed a breakthrough series quality improvement collaborative (BTSC) delivered through a series of learning events. These included a quality improvement curriculum in which rapid tests of change were designed, performed and shared by contributing teams, with the goal of altering dialysis unit organisation and culture to facilitate the implementation of SHC [11]. The objective of the study presented here was to test this complex, multi-centre intervention. To do so we designed a stepped wedge cluster randomised trial (SWCRT) of SHC delivered through a BTSC (SHAREHD) conducted across twelve renal centres with the primary endpoint being to increase the proportion of patients doing five or more tasks or undertaking HD at home [12,13].

## Materials and methods

### Study design

The trial design was an 18 month closed cohort stepped wedge cluster randomised trial (SWCRT) conducted in 12 renal centres (units of cluster randomisation) in England, according to a published protocol consistent with the extension to cluster randomized trials of the Consolidated Standards Of Reporting [13]. It ran from October 2016 to March 2018: following a control period of 6 months, the first group of 6 centres received the intervention and 6 months later the 6 remaining centres received it (Figs 1 and 2). The study adhered to the declaration of Helsinki, ethical approval was obtained from West London & GTAC Research Ethics Committee (IRAS project ID 212395). We focused on undertaking many of the other activities involved in setting up the breakthrough series collaborative, engaging with the teams from the centres and planning the various workstreams, and as a result the trial registration (ISRCTN Number 93999549) was delayed until after the first patient had been consented. The authors confirm that all ongoing and related trials for this intervention by this investigatory group are registered. The protocol was published before the study was completed and before any results were analysed [12].

### Participants

The 12 renal centres had responsibility for the local organisation and delivery of HD treatment and were invited to participate by the chief investigator (MEW) in October 2015. Between October 2016 and February 2017 prevalent HD patients treated at these centres were approached by research nurses, and written, informed consent was obtained to participate in a questionnaire-based study. The specific details of the SWCRT and breakthrough series collaborative were not shared with the patients.

Inclusion criteria: Cluster level: English renal centres with a hospital or satellite-based hemodialysis programme. Patient level: patients to be established on centre-based hemodialysis and have capacity to give written informed consent. Exclusion criteria were those who are too unwell to engage in the study, as judged by the clinical team, or unable to understand written and verbal communication in English.

Centres participating in the trial were: Sheffield Teaching Hospital NHS Foundation Trust, Central Manchester Healthcare Trust, City Hospitals Sunderland NHS Foundation Trust, East & North Hertfordshire NHS Trust, Guy's & St Thomas NHS Foundation Trust, Heart of England Foundation Trust, Leeds teaching Hospitals NHS Trust, The Royal Wolverhampton NHS Trust, North Bristol NHS Trust, University Hospital of North Midlands NHS Trust, Nottingham University Hospitals NHS Trust

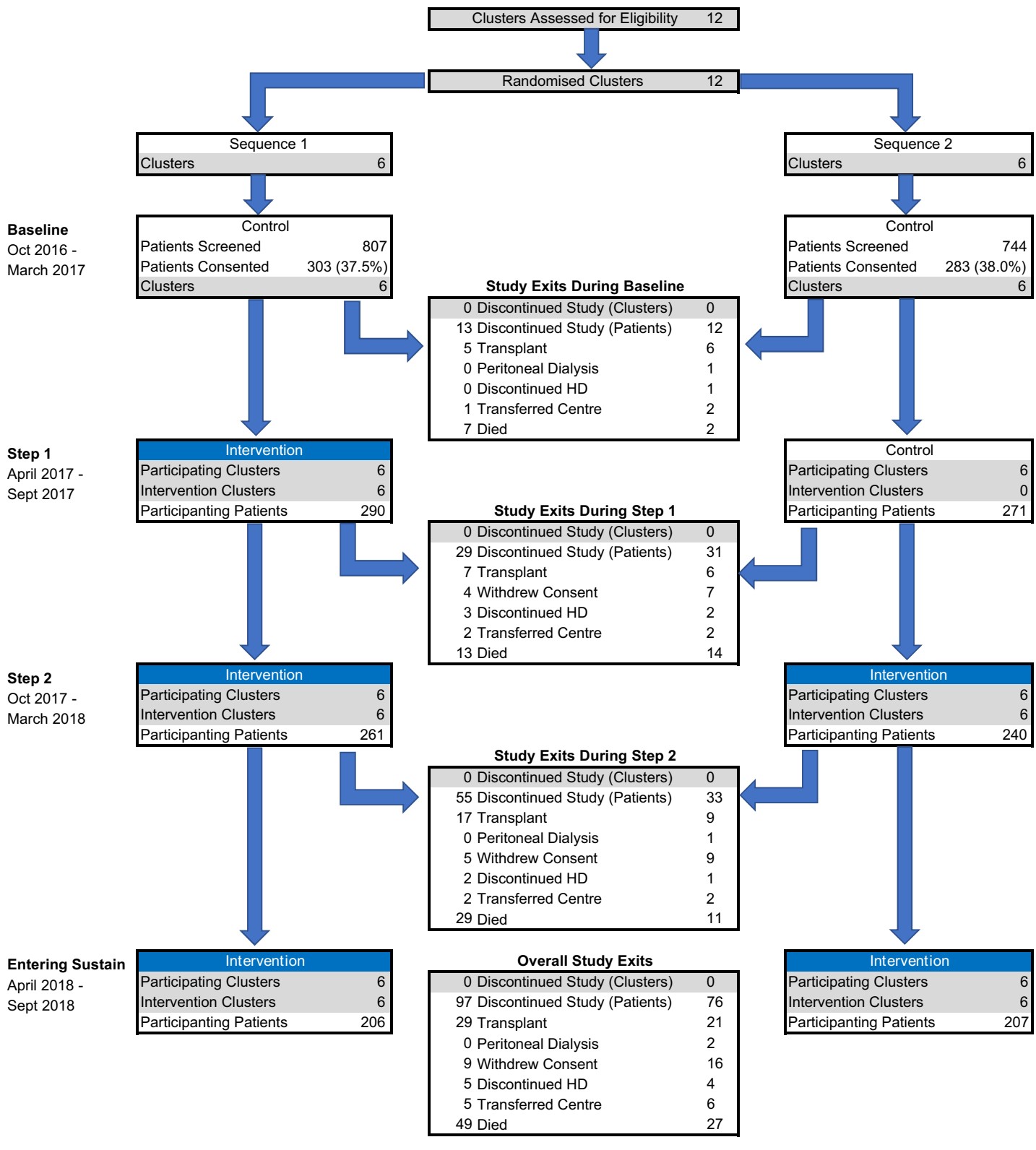

**Fig 1. Study patient and cluster flow though the stepped wedge randomised controlled trial.**

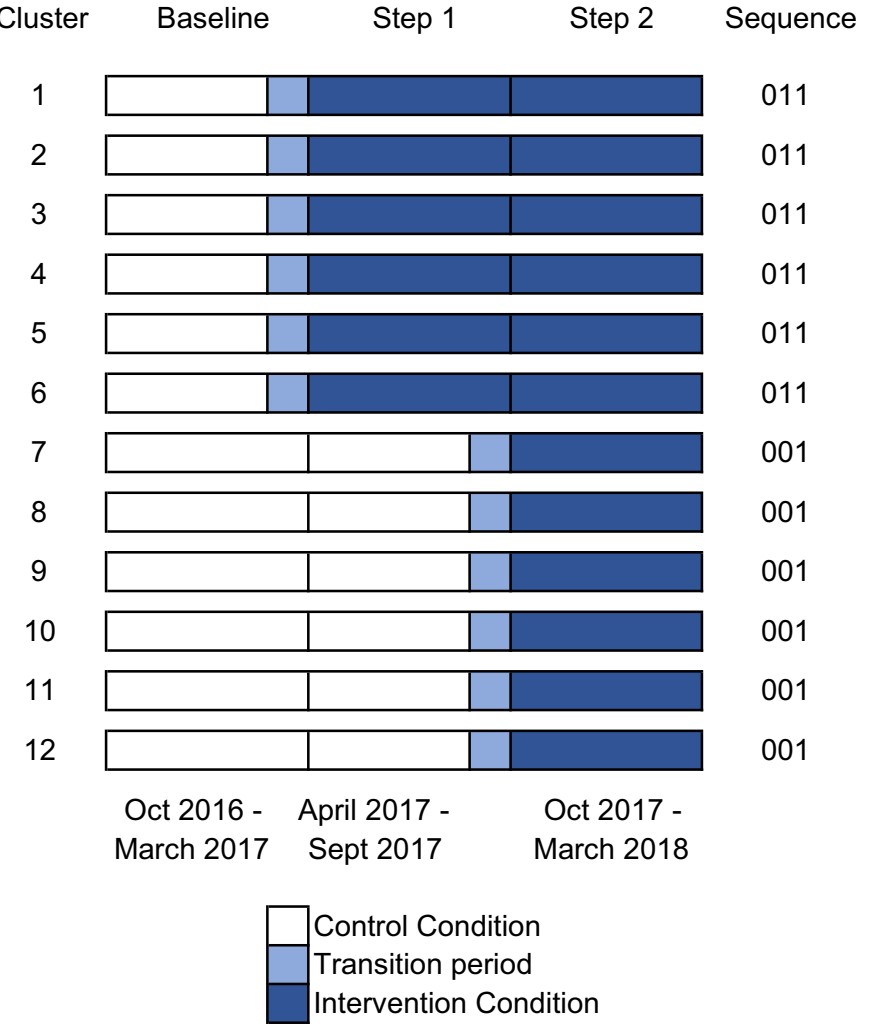

**Fig 2. Diagram of stepped intervention across 12 clusters (renal centres).**

## Randomisation and masking

In July 2016 the study statistician (SJW) used computer generated numbers to produce an allocation sequence for the 12 dialysis centre clusters (6 early and 6 late). The study manager (SL) then informed participating centres of their individual sequence after they had agreed to participate in the trial. There was no patient-level stratification and neither clusters or patients were blinded to the sequence allocation.

## Procedures

The SHAREHD Intervention was delivered through a BTSC [14] involving implementation teams from participating centres made up of approximately five individuals from each site including nursing staff, clinicians, patient partners and additional personnel (e.g. psychologist, service managers as determined by individual sites). Components of the collaborative are presented in Fig 3 with data in S1 Table. Teams met every six weeks at learning events designed to enable adoption of SHC through sharing patient and clinician experiences, teaching and reviewing improvement methodologies and designing PDSA cycles. Progress and outcome of

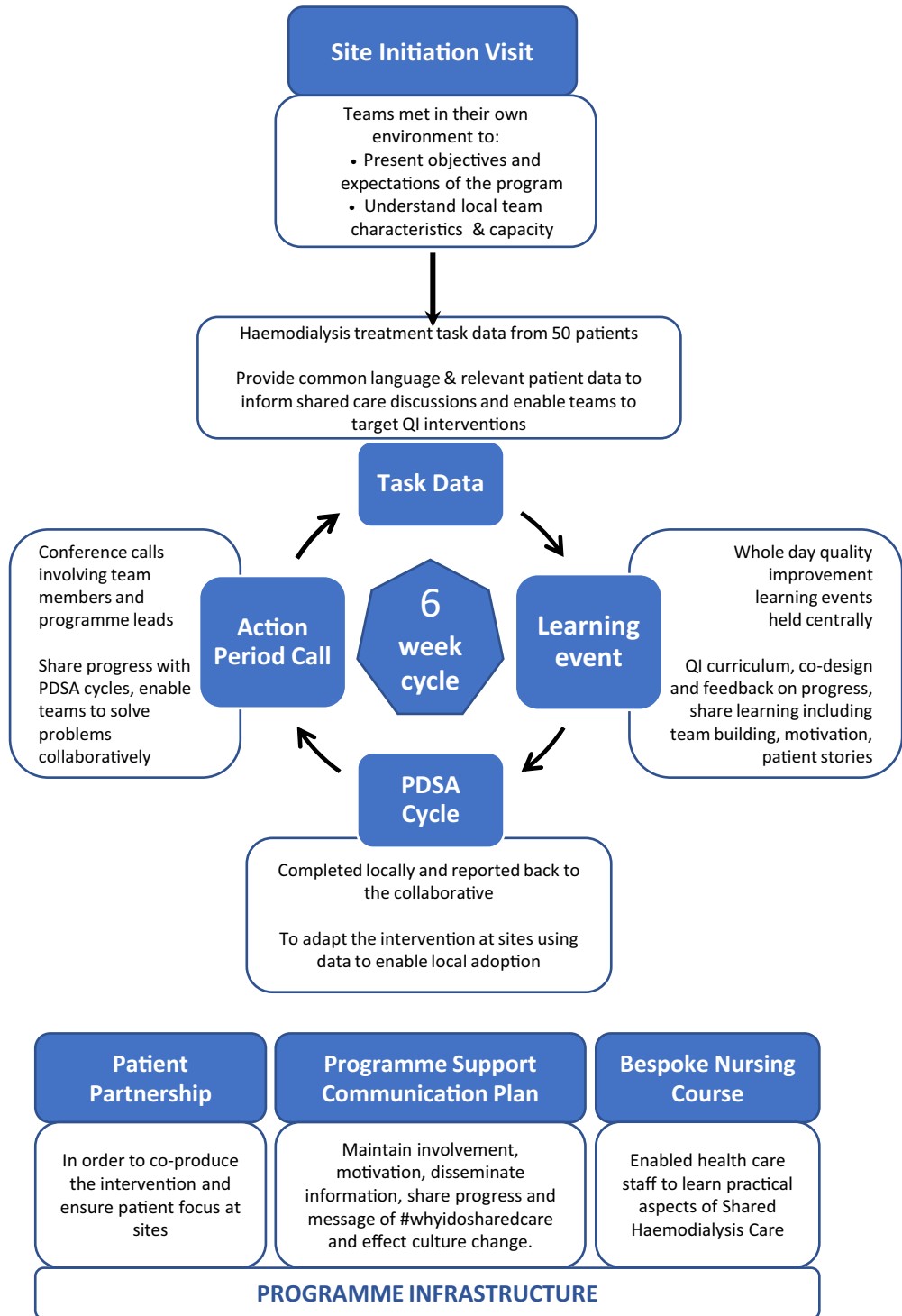

**Fig 3. Diagram of the SHAREHD intervention.**

these PDSA cycles was informed by real-time task data from participating patients and reviewed at subsequent action period calls. The programme was led by a clinician (MEW), programme manager (SL), patient representative (AH), rapid data analytics (JF) and

developmental evaluation of both the programme and the implementation at sites (SA). The program was supported by bespoke materials (patient information leaflets, training manuals and an adoption roadmap available via the SHC website) [15], social media (e.g. Facebook, WhatsApp and @sharemydialysis on Twitter) and newsletters. An established SHC course for health care staff was available and utilised by teams as required [16].

The BTSC was aligned with the SWCRT design. Two sequences of learning events each containing teams from six participating sites were undertaken. The first sequence attended the initial 4 events over six months. Sequence 2 then had a single event to learn key concepts followed by 4 further events in combination with the sequence 1 sites (Figure in S1 Fig) over 6 months. Sites attended events and were instructed to apply their learning during transition or intervention periods only.

Data from patients participating in the research study including endpoints were collected using research nurse and self-completed paper instruments at three-monthly intervals for HD tasks and six-monthly intervals for secondary endpoints with the exception of hospitalisation which was obtained through data linkage to hospital episode statistics by the National Health Service (NHS) Digital Data Access Request Service [17].

## Outcomes

The primary binary outcome was a change in the proportion of participating hemodialysis patients completing 5 or more out of 14 tasks (independently or supervised) or transferring to HHD, sampled at three monthly intervals throughout the study. We selected 5 tasks as the primary outcome measure based on QI work conducted in Yorkshire and the Humber (UK) which suggested that this number indicated patient engagement in the dialysis process through conducting tasks beyond handwashing and performing observations (Table 1) [18]. The secondary binary outcome was an absolute increase in HHD and in-centre independent dialysis. These endpoints were collected towards the end of each three-month period in order to allow patients time to have been exposed to the intervention (Fig 2). Where observations were obtained during the transition period between control and intervention these were assigned to the control.

The secondary endpoint of HHD or independent (completely self-caring) in-centre hemodialysis was defined as the move to HHD or the tasks asterixed in Table 1. The secondary endpoints of change in patient activation; [4,19] quality of life (EQ-5D-5L) [20] and POS-S Renal symptom score components (anxiety, depression and pain) [21] were measured six-monthly at pre-defined timepoints. Hospitalisation was used to assess safety and adverse events, and was assessed as change in all-cause and cause specific–dialysis access, fluid overload and cardiovascular events based on ICD10 codes (data in S2 Table).

## Statistical analysis

Using a recommended ICC value of 0.05 [22], A SWCRT design of 3 steps (including baseline) and 12 clusters of 25 patients, with 6 clusters randomised at each step, would have a 90% power to detect an increase in event rate from 15% to 30% [23] as statistically significant at the 5% two-sided level. Assuming the baseline independent in-centre HD rate was around 2% in participating clusters with the same assumptions as the primary endpoint the SWCRT design has 80% power to detect an increase in the event rate from 2% to 7.2% as statistically significant at the 5% two-sided level. In recognition of the background mortality and renal transplantation rate and to mitigate the risk of incomplete data collection, the target recruitment per participating site was doubled to 50.

Absolute changes in primary and secondary endpoints were assessed comparing the first available observation for all patients during the baseline period with the last available

observation for all patients during the second step. The categorical primary endpoints of five tasks or home HD and secondary endpoints of anxiety, depression and pain (absent or mild compared to moderate, severe or overwhelming due to their distribution) and hospitalisation per three-month period were analysed using mixed effects logistic regression performed on all observations and including the stepped wedge randomly assigned exposure to the control or intervention at the time of data collection (intention to treat). Continuous endpoints including numbers of tasks, patient activation measure score and EQ5D utility were analysed using mixed effects linear regression adopting the same approach as above. Primary and secondary outcomes were evaluated accounting for clustering within participating renal centres using a random intercept (a centre-specific baseline proportion or an endpoint), and within patient using a random intercept (a patient-specific baseline proportion of an endpoint). All multivariable models were adjusted for the baseline variables of age (<35, 35–49, 50–64, 65–79, 80+), gender, time on dialysis (years), marital status, health literacy (adequate defined as "somewhat" or better to the question "How confident are you filling out medical forms by yourself?" [24]), EQ5D utility value and comorbid score derived from linked administrative data [25]. Baseline EQ5D utility score was included in all models except when it was analysed as secondary outcome measure. Endpoints are reported with and without adjustment for time (measured in months from beginning of baseline period) to account for any underlying secular trends in endpoints [13]. A patient was censored from primary and secondary endpoint analyses if they were transplanted, switched to peritoneal dialysis, withdrew consent, moved centre, discontinued dialysis or died. Incident dialysis patients were defined as receiving dialysis for less than six months and prevalent longer than six months at study inception.

Prespecified sensitivity analyses exploring the effect of the intervention on subgroups of individuals were defined in our analysis plan and reviewed by our evaluation advisory board. All analyses were performed on STATA version 14.2 (StataCorp. 2015. College Station, TX).

## Results

The 12 participating renal centres were identified and agreed to participate between June and September 2015. Between October 2016 and January 2017 1551 patients were screened across the 12 centres and 586 patients consented to participate (303 in the 6 centres that began the intervention January 2017 and 283 patients in the 6 centres that began the intervention in July 2017, Figs 1 and 2). There were no deviations from the scheduled steps and no clusters were lost during the trial. The fidelity of the intervention was evaluated based on the conduct and participation in learning events and action period calls, PDSA cycles and task data collection (data in S1 Text).

### Patient characteristics and numbers analysed

The baseline characteristics of the patients recruited into the trial are detailed in Table 2, stratified by the randomisation and their baseline primary endpoint status. Patient characteristics by cluster are available in data in S4 Table. The flow of patients and renal centres through the two stepped sequences is illustrated in Fig 1, showing that by the end of the 18 month study 173 patients (29.5%) of patients had discontinued the study, however missing data for HD tasks (n = 8) and adjustment variables (n = 179) resulted in the exclusion of 187 patients from the multi-variable logistic regression model assessing the primary endpoint of proportion undertaking 5 or more HD tasks or home hemodialysis.

### Outcomes and estimation

Of the 449 patients who had their tasks measured during both baseline and intervention periods, the number undertaking five or more tasks independently or supervised increased from

**Table 2. Baseline patient characteristics.**

| | Randomisation | | Baseline Ind/Sup Tasks | | Observed values |
|---|---|---|---|---|---|
| | Sequence 1 | Sequence 2 | <5 | > = 5 | |
| n | 303 | 283 | 321 | 253 | |
| Age (years) | 62.8 (15.6) | 62.9 (15.6) | 65.8 (15.6) | 59.2 (14.8) | 547 |
| Time since first dialysis (years) | 5.1 (5.4) | 5.8 (8) | 4.3 (4.9) | 6.7 (8.1) | 498 |
| Gender (Male) | 179 (60.9%) | 148 (62.2%) | 174 (60.2%) | 152 (62.8%) | 532 |
| Ethnicity (Caucasian) | 250 (85.6%) | 180 (76.9%) | 242 (84.6%) | 187 (78.2%) | 526 |
| Comorbidities | | | | | 520 |
| Chronic Obstructive Pulmonary Dis. | 25 (8.9%) | 27 (10.8%) | 32 (10.9%) | 19 (8.4%) | |
| Heart Failure | 51 (18.2%) | 51 (20.3%) | 65 (22.1%) | 36 (15.9%) | |
| Cerebrovascular Disease | 20 (7.1%) | 25 (10.0%) | 33 (11.2%) | 12 (5.3%) | |
| Previous Myocardial Infarction | 59 (21%) | 43 (17.1%) | 65 (22.1%) | 36 (15.9%) | |
| Lymphoma | 3 (1%) | 5 (2.0%) | 4 (1.4%) | 3 (1.3%) | |
| Neurological Disease | 16 (5.7%) | 9 (3.6%) | 17 (5.8%) | 8 (3.5%) | |
| Vascular Procedure | 8 (2.9%) | 9 (3.6%) | 13 (4.4%) | 4 (1.8%) | |
| Valvular Heart Disease | 44 (15.7%) | 38 (15.1%) | 51 (17.4%) | 29 (12.8%) | |
| Cancer | 24 (8.5%) | 20 (8.0%) | 21 (7.1%) | 21 (9.3%) | |
| Connective Tissue Disease | 12 (4.3%) | 15 (6.0%) | 15 (5.1%) | 12 (5.3%) | |
| Diabetes | 112 (39.9%) | 104 (41.4%) | 129 (43.9%) | 84 (37.2%) | |
| Comorbid Score | 1.51 (1.38) | 1.55 (1.42) | 1.71 (1.45) | 1.33 (1.31) | 520 |
| EQ5D Utility | 0.68 (0.27) | 0.71 (0.26) | 0.67 (0.28) | 0.74 (0.23) | 485 |
| PAM score | 56.2 (18.6) | 58.3 (19) | 50.8 (15.6) | 64.3 (19.6) | 456 |
| PAM level | | | | | 456 |
| 1—Passive & Overwhelmed | 80 (31.5%) | 50 (24.8%) | 93 (38.3%) | 37 (17.4%) | |
| 2—Lack Knowledge & Confidence | 57 (22.4%) | 38 (18.8%) | 66 (27.2%) | 29 (13.6%) | |
| 3—Taking Action | 75 (29.5%) | 70 (34.7%) | 66 (27.2%) | 79 (37.1%) | |
| 4—Adopted behaviours | 42 (16.5%) | 44 (21.8%) | 18 (7.4%) | 64 (31.9%) | |
| Anxiety (moderate or worse) | 74 (28.1%) | 55 (24.4%) | 64 (24.8%) | 65 (28.3%) | 488 |
| Depression (moderate or worse) | 73 (27.9%) | 41 (18.2%) | 61 (23.7%) | 53 (23.0%) | 488 |
| Pain (moderate or worse) | 104 (39.5%) | 84 (37.8%) | 105 (40.9%) | 83 (36.4%) | 487 |
| Poor Mobility (moderately impaired or worse) | 152 (57.4%) | 99 (44.0%) | 149 (57.1%) | 102 (44.5%) | 490 |
| Limited Health Literacy | 73 (28.0%) | 54 (24.4%) | 85 (31.0%) | 42 (17.7%) | 482 |
| Education (no formal qualification) | 108 (37.8%) | 71 (30.0%) | 107 (38.1%) | 71 (29.6%) | 522 |
| Number of tasks (mean) | | | | | |
| Independent or Supervised | 5 (4.1) | 5.1 (3.9) | 2.3 (1.3) | 8.6 (3.4) | 574 |
| Independent | 4.3 (3.8) | 3.9 (3.3) | 2 (1.2) | 6.9 (3.6) | 574 |
| Interest in Home HD | | | | | 546 |
| Yes | 40 (13.9%) | 41 (15.9%) | 33 (10.7%) | 48 (20.3%) | |
| No | 202 (70.1%) | 179 (69.4%) | 234 (75.7%) | 147 (62.0%) | |
| Maybe | 46 (16.0%) | 38 (14.7%) | 42 (13.6%) | 42 (17.7%) | |
| Self-needling interest (probably do it or better) | 84 (35.9%) | 61 (32.6%) | 50 (23.0%) | 95 (46.6%) | 421 |

Full list of POS-S symptoms available in data in S3 Table.

205 to 244 following the intervention (45.6% vs 52.3%, absolute change 6.2%, P = 0.010, 95% CI: 1.4 to 11.0%).

The trend in patients performing five or more tasks or undertaking home hemodialysis is shown in Fig 4A, stratified by randomised sequence. The time adjusted odds ratio for the

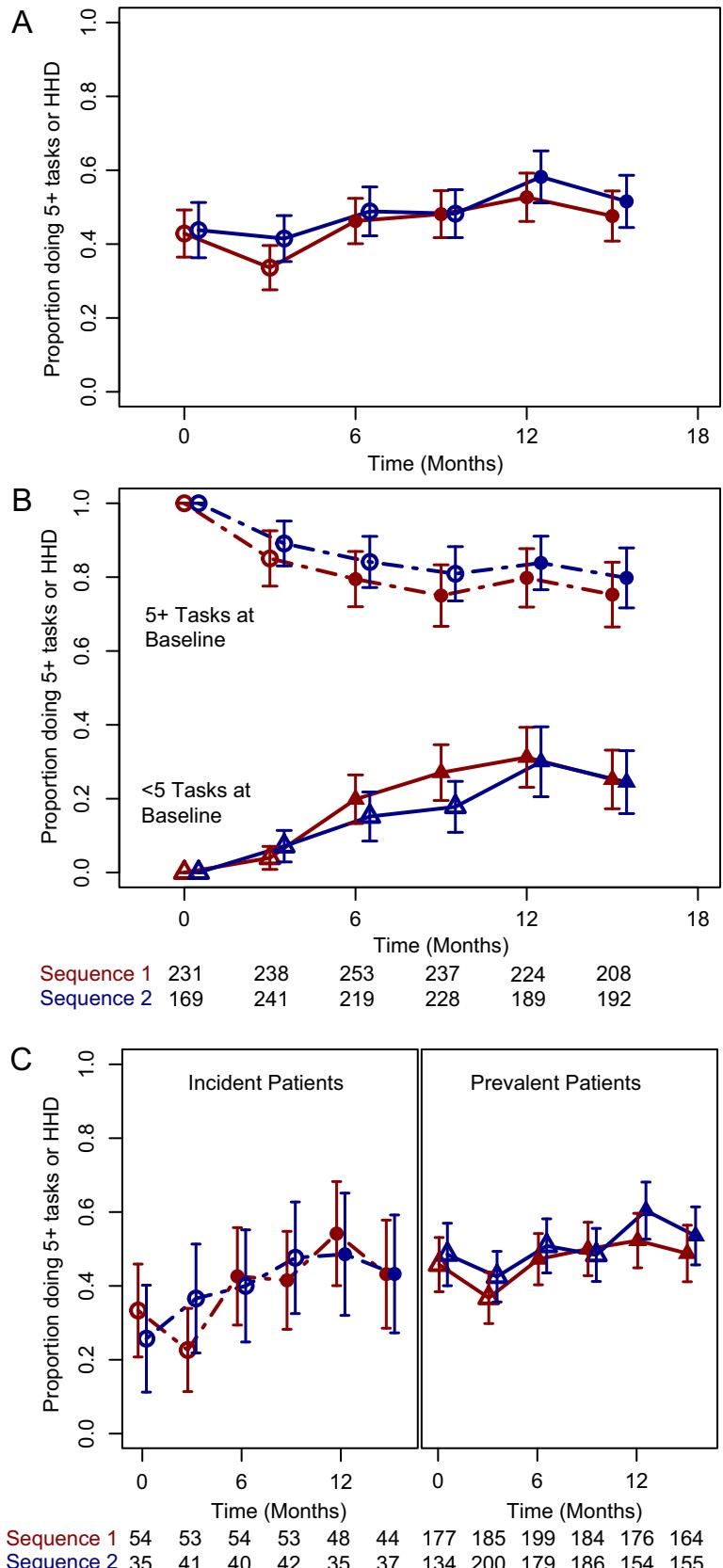

**Fig 4. Secular trend of primary endpoint (5 or more tasks or home hemodialysis).** (A) Overall, according to sequence of randomisation (B) Secular trend of the primary endpoint stratified by number of tasks at baseline. (C) Secular trend of the primary endpoint stratified by incident (within 6 months of starting hemodialysis) or prevalent (6 months or longer on hemodialysis).

intervention for undertaking five or more tasks independently or supervised or home hemodialysis was 1.62 (95% CI 1.02–2.60, P = 0.043) in analyses not including baseline variables and 1.63 (95% CI 0.94–2.81, P = 0.080) in the analysis adjusted for baseline covariates (Table 3). Changes in individual tasks are shown in Fig 5, showing improvements in the proportion doing both supervised and unsupervised tasks, and are reported stratified by number of baseline tasks in supplementary figures (figure in S2 Fig, figure in S3 Fig).

The proportion of patients performing their dialysis independently in centre was 5.2% (21/402) at the end of the baseline period and 6.9% (28/402) at the end of the study (absolute difference 1.7%, 95%CI -1.0 to 4.5%) and 21 patients (5.0% of 402 completing the study) moved from in-centre to home HD. The overall improvement of this combined endpoint was from 7.6% to 11.6% (difference 4.0%, 95% CI 1.0% to 7.0%), however the relatively small number of events precluded the use of a multi-level multivariable model on this endpoint.

**Table 3. Primary endpoint of five or more tasks or home hemodialysis, and secondary endpoints: Effect sizes.**

| Endpoint Analysis | Time (per month) | Effect Size of Intervention (95% CI) | P | N | ICC centre | ICC patient |
|---|---|---|---|---|---|---|
| | | **PRIMARY ENDPOINT** | | | | |
| Primary—5+ Tasks or HHD | | | | | | |
| Absolute Proportion doing tasks | | 45.6% (205/449) vs 52.3% (244/449) Difference 6.2% (1.4–11.0) | 0.01 | | | |
| Crude OR | 1.00 (0.96–1.05) | 1.62 (1.02–2.60) | 0.043 | 578 | 0.237 | 0.818 |
| Multivarable adjusted OR | 1.01 (0.95–1.05) | 1.63 (0.94–2.81) | 0.08 | 399 | 0.220 | 0.762 |
| Crude OR without time | - | 1.68 (1.28–2.21) | <0.001 | 578 | 0.237 | 0.818 |
| Multivarable adjusted OR without time | - | 1.59 (1.16–2.19) | 0.004 | 399 | 0.220 | 0.762 |
| | | **SECONDARY ENDPOINTS** | | | | |
| Independent ICHD or HHD | | 7.5% (32/423) vs 11.6% (49/423) Difference 4.0 (1.0–7.0) | 0.008 | | | |
| Number of tasks (Independent or Supervised) | 0.01 (-0.05–0.08) | 0.31 (-0.26–0.89) | 0.283 | 399 | 0.179 | |
| Number of tasks (Independent) | 0.05 (-0.01–0.10) | 0.21 (-0.31–0.72) | 0.43 | 399 | 0.170 | |
| Patient Activation Score (0–100) | -0.12 (-0.53–0.28) | 1.26 (-2.88–5.41) | 0.551 | 393 | 0.063 | |
| EQ5D Utility Value (0: Dead, 1: Perfect Health) | 0.0 (-0.01–0.00) | 0.01 (-0.06–0.07) | 0.806 | 409 | 0.049 | |
| Depression (Moderate or worse, OR) | 1.01 (0.93–1.10) | 1.05 (0.46–2.40) | 0.916 | 390 | 0.006 | 0.513 |
| Anxiety (Moderate or worse, OR) | 1.00 (0.91–1.08) | 0.90 (0.43–1.90) | 0.787 | 390 | 0.006 | 0.464 |
| Pain (Moderate or worse, OR) | 0.96 (0.90–1.03) | 1.89 (0.99–3.61) | 0.054 | 390 | 0.000 | 0.359 |
| All Cause Hospitalisation OR | 1.00 (0.97–1.04) | 1.00 (0.68–1.47) | 1.000 | 399 | 0.007 | 0.170 |
| Infection Hospitalisation OR | 1.01 (0.95–1.08) | 1.15 (0.62–2.11) | 0.662 | 399 | 0.000 | 0.063 |
| Fluid Overload Hospitalisation OR | 1.10 (0.98–1.23) | 0.24 (0.07–0.80) | 0.019 | 399 | 0.065 | 0.065 |
| Vascular Access Hospitalisation OR | 1.08 (1.00–1.17) | 0.78 (0.35–1.75) | 0.551 | 399 | 0.000 | 0.032 |
| Emergency Room Attendance OR | 1.02 (0.98–1.05) | 1.00 (0.70–1.42) | 0.985 | 399 | 0.000 | 0.176 |

OR: Odds Ratio. Mixed effects logistic or linear regression model with a random effects (random intercept for cluster and participant where ICC patient quoted) and fixed effects for intervention and time and baseline covariates. Adjusted for the baseline variables of age (categories), gender, time on dialysis (years), marital status, health literacy (adequate or inadequate), EQ5D utility value and comorbid score (derived from chronic obstructive pulmonary disease, congestive cardiac failure, cerebrovascular accident, acute myocardial infarction, neurological disease, vascular intervention, valvular heart disease, cancer, connective tissue disease and diabetes).

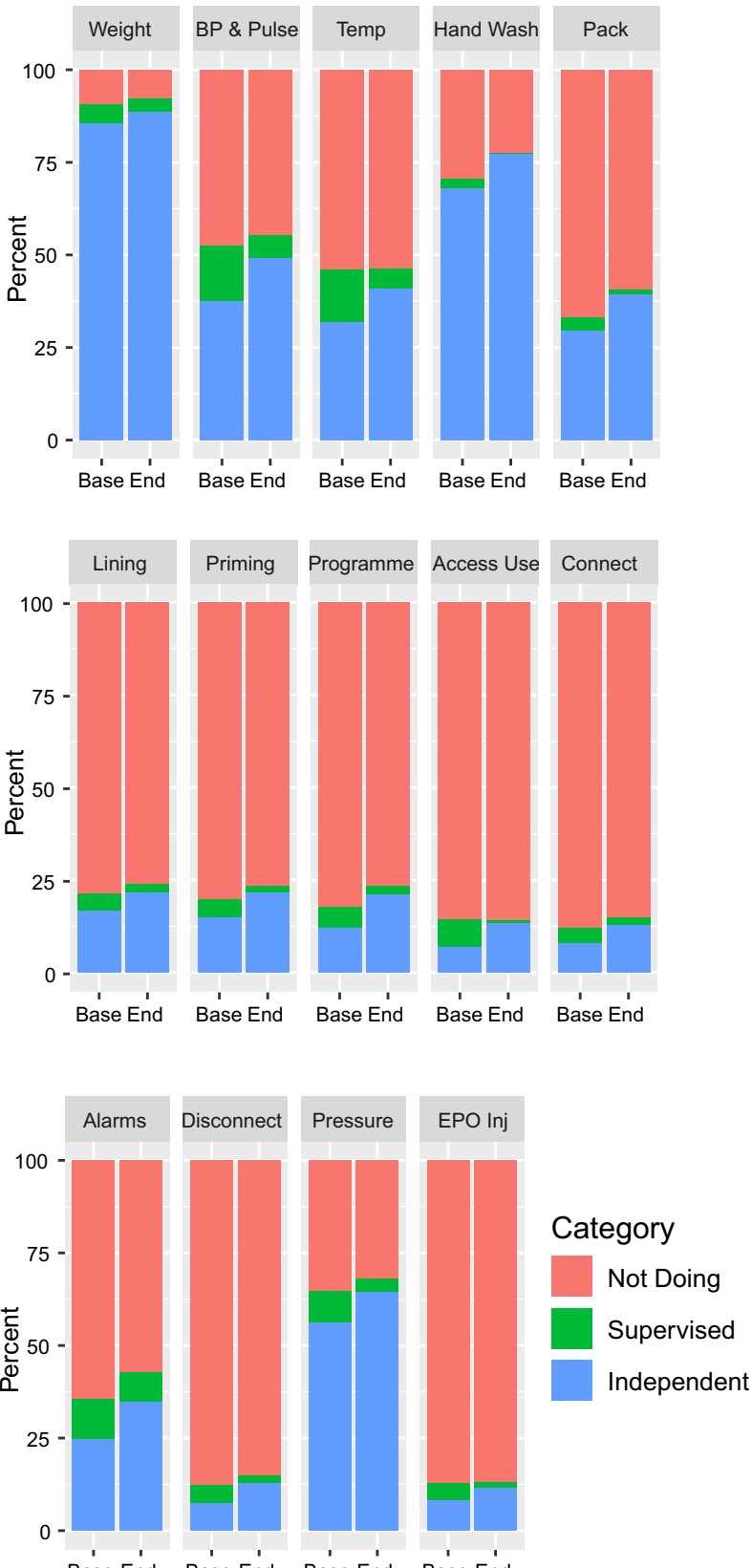

**Fig 5. Patient participation in individual dialysis tasks at baseline and end of the stepped wedge randomised controlled trial in individuals with data at both timepoints (n = 427).** Categories: Independent (blue), Supervised (green), and Not Doing (red). Figure stratified by baseline participation: Figure in S2 Fig and figure in S3 Fig.

There was no statistically significant impact of the stepped intervention on the patient activation measure® (adjusted mean difference 1.26, 95% CI -2.88 to 5.41, P = 0.551), EQ5D quality of life (adjusted mean difference 0.01, 95% CI -0.06 to 0.07, P = 0.806), number of HD tasks, or symptoms of depression, anxiety or pain (Table 3, figure in S4 Fig). There was no significant difference in hospitalisation according to exposure to the intervention (Table 3, data in S6 Table, figure in S5 Fig).

## Subgroup and sensitivity analyses

Stratifying by the primary endpoint of five or more tasks at baseline showed differing effects of the intervention. 80.0% (123/205, 95% CI 74.7–85.7%, P<0.001) of patients who began the study doing five or more tasks were still doing five or more at the end of the study, whereas 28.3% (69/244, 95% CI 22.2–34.3, P<0.001) who began the study doing less than five tasks were doing more than five tasks at the end of the study (Fig 4B). The time and multi-variable adjusted odds ratio effect of the intervention in patients doing less than 5 tasks at baseline was 3.71 (95% CI 1.66–8.31, P = 0.024, data in S5 Table). Having removed time from the endpoint models the odds ratio for patients completing five or more tasks or performing home hemodialysis was 1.59 (95% CI 1.16–2.19, P = 0.004, Table 3).

## Discussion

This 12-site stepped wedge cluster randomised trial (SWCRT) evaluating a breakthrough series collaborative (BTSC) supporting patients to learn treatment related tasks significantly improved the absolute proportion of patients undertaking the combined end-point of 5 of more HD tasks or home HD, however the adjusted odds ratio for the intervention was not significant. Significant increases in the combined end-point of dialyzing independent in-centre and home HD were observed, as was the increase in participation in HD tasks in patients who were performing fewer than 5 dialysis tasks at baseline. However, the secondary endpoints of patient activation, EQ5D quality of life and hospitalisation were unaltered.

Demonstrating the impact of quality interventions in kidney disease is challenging, however there is evidence for the use of quality improvement collaboratives in renal replacement therapy [26], particularly around the reduction of central venous catheter infection rates and peritoneal infection rates but this approach has not been used to deliver self-management support in dialysis [27], Despite associations between improved health-related quality of life and HHD and the SHC intervention being effective in promoting independent and home HD, gains in health-related quality of life not observed. Our study did not identify increased infection- and dialysis-access-related hospitalisations associated with performing more tasks, unlike observational data from the US showing increases in these events in home- compared to in-centre patients [28]. The observed reduction in fluid overload admissions may be a consequence of patients performing their own weight and programming their HD machines leading to improved knowledge around fluid management [29].

The strengths of this study include that it was multi-centre and of relatively large size and that groups were well randomised at baseline. In order to maintain external generalizability inclusion criteria were broad and consented patients were representative of the prevalent HD population including the multi-morbidity associated with this group. The BTSC and associated co-production with service users resulted in adaption of the intervention at participating

sites to take account of contextual issues which can impact on intervention efficacy [30]. The SWCRT design enabled cost-effective evaluation of the impact of this complex intervention on the longitudinal changes in endpoints using a closed design [26]. However, a weakness of the that design was that all patients received the intervention during the final phase when they were potentially at their frailest due to the progressive impact of their medical condition. This increasing frailty may have contributed to a reduced dialysis task participation during the course of the study among those who were undertaking more than 5 tasks at the outset, and baseline frailty could have prevent a subgroup of individuals with low task participation at baseline from increasing their task participation. The combination of these factors and the inclusion of time in the outcome models may have resulted in an under estimation of the effect of the intervention, and for this reason and in line with guidance we report our models both with and without time included [13,31]. Other weaknesses included missing data reducing the sample size for multivariable models, a higher than expect baseline rate of dialysis-related tasks (45.6 observed vs 15% assumed) and centre interclass correlation (observed 0.179 vs 0.05 assumed), the unblinded assessment of patient tasks and that the act of collecting task data led to greater engagement during control periods.

Healthcare providers intending to increase home dialysis use may consider the SHC intervention since this study demonstrated impact on independent and home HD use. The intervention had the greatest impact on individuals who were undertaking fewer tasks at baseline and possibly those with the lowest levels of patient activation. However, as individuals performing fewer tasks at baseline appear to be more comorbid if they increased their number of tasks are a result of the intervention, they could subsequently reduce their number of tasks due to this frailty. Future interventions in this area should explore approaches to rehabilitate individuals whose self-efficacy has acutely or chronically declined, and intervention/trial designs that focus on incident patients and utilise Transitional Care Units to support individuals who are starting dialysis to learn tasks [32].

In conclusion, despite the difficulties of studying a prevalent, highly co-morbid dialysis population, the delivery of a break-through series collaborative designed to support greater patient participation in centre-based HD was safe and effective at improving the number of individuals performing dialysis independently or at home, and increased HD tasks in-centre among patients who were performing less than five. Recognition of the impact of this intervention while acknowledging the tendency for patients to become frailer over time are important considerations when responding to HD policy recommendations designed to increase self-management.

## Supporting information

**S1 Checklist.**
(PDF)

**S1 Fig. Sequence of learning events and action period calls.**
(PDF)

**S2 Fig. Patient participation in individual dialysis tasks at baseline and end of the stepped wedge (<5 tasks at baseline).**
(PDF)

**S3 Fig. Patient participation in individual dialysis tasks at baseline and end of the stepped wedge (5+ tasks at baseline).**
(PDF)

**S4 Fig. Secular trends in secondary endpoints, stratified by randomisation sequence.**
(PDF)

**S5 Fig. Secular trends in hospitalisation.**
(PDF)

**S1 Table. Components of the breakthrough series collaborative underpinning the delivery of the intervention.**
(PDF)

**S2 Table. ICD10 codes for the reasons for hospitalisation.**
(PDF)

**S3 Table. Individual symptoms of the POS-S renal score.**
(PDF)

**S4 Table. Cluster characteristics.**
(PDF)

**S5 Table. Subgroup analysis of primary endpoint.**
(PDF)

**S6 Table. Hospitalisation rates, rate ratios and odds ratios.**
(PDF)

**S1 Text. Description and results of fidelity assessment.**
(PDF)

**S1 File.**
(PDF)

## Acknowledgments

The study team wish to acknowledge and thank the following contributing team members: Site principle investigators: Veena Reddy: Sheffield Teaching Hospital NHS Foundation Trust; Sandip Mitra: Central Manchester Healthcare Trust; Saeed Ahmed: City Hospitals Sunderland NHS Foundation Trust; Paul Warwicker: East & North Hertfordshire NHS Trust; Nicola Kumar: Guy's & St Thomas NHS Foundation Trust; Joyti Baharani: Heart of England Foundation Trust; Elizabeth Garthwaite: Leeds teaching Hospitals NHS Trust, Babu Ramakrishna: The Royal Wolverhampton NHS Trust, Albert Power: North Bristol NHS Trust; Mark Lambie: University Hospital of North Midlands NHS Trust; Alastair Ferraro: Nottingham University Hospitals NHS Trust; Implementation and research team members: Joanna Blackburn (qualitative research): Barnsley Hospital NHS Foundation Trust; Paul Harriman (quality improvement), Megan Bennett and Richard Simmonds (administrative support); Catherine Stannard & George Swinnerton (Think Kidneys) for processing the Your Health Survey; Sheffield Teaching Hospitals NHS Foundation Trust (Sponsor); Strategic advice from Michael Nation: Kidney Research UK. Prof Sue Mawson for chairing the evaluation advisory board. NIHR CRN research nurses at participating sites for consenting patients and supporting questionnaire completion.

## Author Contributions

**Conceptualization:** James Fotheringham, Tania Barnes, Steven Ariss, Stephen J. Walters, Paul Laboi, Andy Henwood, Rachel Gair, Martin Wilkie.

**Data curation:** James Fotheringham, Louese Dunn, Tracey Young, Stephen J. Walters, Martin Wilkie.

**Formal analysis:** James Fotheringham, Steven Ariss, Stephen J. Walters, Martin Wilkie.

**Funding acquisition:** James Fotheringham, Tania Barnes, Steven Ariss, Stephen J. Walters, Paul Laboi, Andy Henwood, Martin Wilkie.

**Investigation:** James Fotheringham, Louese Dunn, Sonia Lee, Steven Ariss, Tracey Young, Stephen J. Walters, Andy Henwood, Martin Wilkie.

**Methodology:** James Fotheringham, Steven Ariss, Tracey Young, Stephen J. Walters, Rachel Gair, Martin Wilkie.

**Project administration:** James Fotheringham, Louese Dunn, Sonia Lee, Andy Henwood, Martin Wilkie.

**Resources:** James Fotheringham, Tania Barnes, Louese Dunn, Sonia Lee, Paul Laboi, Andy Henwood, Rachel Gair, Martin Wilkie.

**Validation:** Stephen J. Walters.

**Visualization:** James Fotheringham.

**Writing – original draft:** James Fotheringham, Tania Barnes, Louese Dunn, Sonia Lee, Steven Ariss, Tracey Young, Stephen J. Walters, Paul Laboi, Andy Henwood, Rachel Gair, Martin Wilkie.

**Writing – review & editing:** James Fotheringham, Tania Barnes, Louese Dunn, Sonia Lee, Steven Ariss, Stephen J. Walters, Martin Wilkie.

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
