## [Decision Letter · Decision Letter 0]

15 Apr 2021

PONE-D-21-06704

A breakthrough series collaborative to increase patient participation with hemodialysis tasks: a stepped wedge cluster randomised controlled trial

PLOS ONE

Dear Dr. Fotheringham,

Thank you for submitting your manuscript to PLOS ONE. After careful consideration, we feel that it has merit but does not fully meet PLOS ONE’s publication criteria as it currently stands. Therefore, we invite you to submit a revised version of the manuscript that addresses the points raised during the review process.

Please address the comments from the reviewers.

We look forward to receiving your revised manuscript.

Kind regards,

Shahrad Taheri

Academic Editor

PLOS ONE

Journal Requirements:

Thank you for submitting your clinical trial to PLOS ONE and for providing the name of the registry and the registration number. The information in the registry entry suggests that your trial was registered after patient recruitment began. PLOS ONE strongly encourages authors to register all trials before recruiting the first participant in a study.

1) your reasons for your delay in registering this study (after enrolment of participants started);

2) confirmation that all related trials are registered by stating: “The authors confirm that all ongoing and related trials for this drug/intervention are registered”.

Thank you for stating in the text of your manuscript "written, informed consent was obtained to participate in a questionnaire-based study". Please also add this information to your ethics statement in the online submission form.

Please provide the full names of the 12 renal centres.

Thank you for stating the following in the Competing Interests section:

JF has received speaker honoraria from Fresenius medical care, and conducts research funded by the National Institute of Health Research (NIHR), Vifor Pharma and Novartis.

MEW has received speaker honoraria Fresenius and Baxter, has acted on an advisory board for Baxter and has conducted research funded by the NIHR.

SJW has received book royalties from Wiley and has received funds from NIHR, the Department of Health and Medical Research Council.

In your Data Availability statement, you have not specified where the minimal data set underlying the results described in your manuscript can be found. PLOS defines a study's minimal data set as the underlying data used to reach the conclusions drawn in the manuscript and any additional data required to replicate the reported study findings in their entirety. All PLOS journals require that the minimal data set be made fully available. For more information about our data policy, please see http://journals.plos.org/plosone/s/data-availability.

We note you have included a table to which you do not refer in the text of your manuscript. Please ensure that you refer to Table 2 in your text; if accepted, production will need this reference to link the reader to the Table.

Please include captions for *all* your Supporting Information files at the end of your manuscript, and update any in-text citations to match accordingly. Please see our Supporting Information guidelines for more information: http://journals.plos.org/plosone/s/supporting-information.

Reviewers' comments:

Reviewer's Responses to Questions

**Comments to the Author**

1. Is the manuscript technically sound, and do the data support the conclusions?

Reviewer #1: Yes

Reviewer #2: Yes

Reviewer #3: Yes

2. Has the statistical analysis been performed appropriately and rigorously? 

Reviewer #1: Yes

Reviewer #2: I Don't Know

Reviewer #3: Yes

3. Have the authors made all data underlying the findings in their manuscript fully available?

Reviewer #1: Yes

Reviewer #2: Yes

Reviewer #3: Yes

4. Is the manuscript presented in an intelligible fashion and written in standard English?

Reviewer #1: Yes

Reviewer #2: Yes

Reviewer #3: Yes

5. Review Comments to the Author

Reviewer #1: This is a model of a stepped wedge trial with appropriate sample size considerations, and careful analysis adjusted for step. Given that it was a negative (primary outcome) study, it is important to point out in the conclusions whether the assumed sample size parameters were realized in the trial.

Reviewer #2: This is a useful piece of work because it looks at an area of dialysis practice that is very topical at present and looks at outcomes that might plausibly be generated by an increase in the proportion of patients involved in their own treatment. It was disappointing that the intervention did not lead to many of the possible positive outcomes, but this is important data to have. It would be interesting to understand if the authors felt that the main reason behind this was the relatively high level of patients already performing some self-care at baseline, or if they anticipated that a longer time period or more intervention might have made a difference, for example.

It would be useful to understand more clearly the nature of the interventions made to increase self-care at the participating centres, which would also indicate what person-time resource was required to deliver the interventions. This would have relevance to the dialysis community, especially given the likely inclusion of shared care HD / Home HD as part of the GIRFT requirements.

Reviewer #3: This was a very important study examining if an educational quality improvement intervention increased the proportion of in-centre hemodialysis patients completing 5 or more out of 14 tasks (independently or supervised) or transferring to home haemodialysis, using stepped wedge cluster randomised trial design. Although the primary endpoint was not achieved, significantly more patients transferred to HHD and there was an increase in the number of patients able to perform complete =/>5 tasks in the subgroup that had performed <5 tasks at baseline.

I have a couple of comments that the authors might want to address.

The authors suggest that increasing frailty among HD patients with time is the likely reason for not achieving primary endpoint. If that is so, how did those who could perform <5 tasks at baseline, presumably frailer or more cognitively impaired at baseline, show significant improvement?

I would like to see more detailed discussion on the putative reasons for not achieving primary outcome and if a different study design would have shown positive results.

6. PLOS authors have the option to publish the peer review history of their article (what does this mean?). If published, this will include your full peer review and any attached files.

Reviewer #1: No

Reviewer #2: No

Reviewer #3: No

---

## [Author Response · Author response to Decision Letter 0]

7 Jun 2021

A fully formatted response is available at the end of our cover letter

Journal Requirements:

2. Thank you for submitting your clinical trial to PLOS ONE and for providing the name of the registry and the registration number. The information in the registry entry suggests that your trial was registered after patient recruitment began. PLOS ONE strongly encourages authors to register all trials before recruiting the first participant in a study.

1) your reasons for your delay in registering this study (after enrolment of participants started);

We have added the statement: 

We focused on undertaking many of the other activities involved in setting up the breakthrough series collaborative, engaging with the teams from the centres and planning the various workstreams, and as a result the trial registration (ISRCTN Number 93999549) was delayed until after the first patient had been consented. 

2) confirmation that all related trials are registered by stating: “The authors confirm that all ongoing and related trials for this drug/intervention are registered”.

We are not undertaking further trials in this area, and could not speak for other groups. Therefore we have stated: 

The authors confirm that all ongoing and related trials for this intervention by this investigatory group are registered.

3. Thank you for stating in the text of your manuscript "written, informed consent was obtained to participate in a questionnaire-based study". Please also add this information to your ethics statement in the online submission form.

 We have added this wording to the Ethics statement in the submission.

4. Please provide the full names of the 12 renal centres.

The 12 centres were listed in the acknowledgements of our original submission. We assume that the journal wants this listed in the main body in addition. We have added this in the cluster inclusion criteria, but wonder if the journal might see this as unnecessary duplication?

Sheffield Teaching Hospital NHS Foundation Trust, Central Manchester Healthcare Trust, City Hospitals Sunderland NHS Foundation Trust, East & North Hertfordshire NHS Trust, Guy’s & St Thomas NHS Foundation Trust, Heart of England Foundation Trust, Leeds teaching Hospitals NHS Trust, The Royal Wolverhampton NHS Trust, North Bristol NHS Trust, University Hospital of North Midlands NHS Trust, Nottingham University Hospitals NHS Trust

JF has received speaker honoraria from Fresenius medical care, and conducts research funded by the National Institute of Health Research (NIHR), Vifor Pharma and Novartis.

MEW has received speaker honoraria Fresenius and Baxter, has acted on an advisory board for Baxter and has conducted research funded by the NIHR.

SJW has received book royalties from Wiley and has received funds from NIHR, the Department of Health and Medical Research Council.

7. We note you have included a table to which you do not refer in the text of your manuscript. Please ensure that you refer to Table 2 in your text; if accepted, production will need this reference to link the reader to the Table.

 Apologies, in the paragraph “Patient characteristics and numbers analysed” the reference to table 3 should have read table 2.

8. Please include captions for *all* your Supporting Information files at the end of your manuscript, and update any in-text citations to match accordingly. Please see our Supporting Information guidelines for more information: http://journals.plos.org/plosone/s/supporting-information.

Our original submission has captions for all supporting information files. We have attempted to align with the above guidance – we have interpreted it as requiring everything to start with “S” and be sequentially numbered.

We have added one reference in response to a reviewers comments and a reference to a data source. One typographical error existed. We have not identified any other issues.

5. Review Comments to the Author

Reviewer #1: This is a model of a stepped wedge trial with appropriate sample size considerations, and careful analysis adjusted for step. Given that it was a negative (primary outcome) study, it is important to point out in the conclusions whether the assumed sample size parameters were realized in the trial.

Thank you. We performed our sample size estimation based on existing literature and best practices in this area. The baseline number of tasks were assumed to be 15%, with 90% power to detect an increase to 30%, however the baseline level in the trial was 45.6% in those who had their tasks measured in both control and intervention period (first paragraph of outcomes and estimation, table 3). We assumed an interclass correlation of 0.05 for tasks but the observed was 0.179 (table 3). We have included reference to these issues in the discussion as follows:

“a higher than expect baseline rate of dialysis-related tasks (45.6 observed vs 15% assumed) and centre interclass correlation (observed 0.179 vs 0.05 assumed,”

Reviewer #2: This is a useful piece of work because it looks at an area of dialysis practice that is very topical at present and looks at outcomes that might plausibly be generated by an increase in the proportion of patients involved in their own treatment. It was disappointing that the intervention did not lead to many of the possible positive outcomes, but this is important data to have. It would be interesting to understand if the authors felt that the main reason behind this was the relatively high level of patients already performing some self-care at baseline, or if they anticipated that a longer time period or more intervention might have made a difference, for example.

Thank you. Certainly we did observe a higher than expected baseline level of task participation and we expand further in our discussion on the size of this. Although we cannot be 100% sure, we doubt longer follow-up would have made a significant difference because the issue around the decline in task participation in the cohort demonstrated in Figure 4B, which may have affected those who had improved their task participation during the study if follow-up was increased.

We have made more specific reference to the second point in our discussion:

“The intervention had greatest impact on individuals who were undertaking fewer tasks at baseline and possibly those with the lowest levels of patient activation although these individuals could regress as was observed in those with higher task participation in baseline.” 

It would be useful to understand more clearly the nature of the interventions made to increase self-care at the participating centres, which would also indicate what person-time resource was required to deliver the interventions. This would have relevance to the dialysis community, especially given the likely inclusion of shared care HD / Home HD as part of the GIRFT requirements.

Thank you for this important question – the person-time resource (specifically nurses) use required to deliver the intervention at sites is the subject of a separate analysis. It can be indirectly estimated from the S2 table. Because of the methods and granularity associated with the comprehensive person-time analysis it cannot be accommodated in this manuscript and will form a separate output.

Reviewer #3: This was a very important study examining if an educational quality improvement intervention increased the proportion of in-centre hemodialysis patients completing 5 or more out of 14 tasks (independently or supervised) or transferring to home haemodialysis, using stepped wedge cluster randomised trial design. Although the primary endpoint was not achieved, significantly more patients transferred to HHD and there was an increase in the number of patients able to perform complete =/>5 tasks in the subgroup that had performed <5 tasks at baseline.

I have a couple of comments that the authors might want to address.

The authors suggest that increasing frailty among HD patients with time is the likely reason for not achieving primary endpoint. If that is so, how did those who could perform <5 tasks at baseline, presumably frailer or more cognitively impaired at baseline, show significant improvement?

Thank you – this issue is important. In our manuscript when we refer to frailty as a mechanism we originally cite it as a potential reason individuals with a higher number of baseline tasks subsequently reduce their number of tasks. One might assume, therefore, that those who do then increase their task participation are from the subgroup of patients who are doing fewer tasks at baseline and are more likely to be those with less comorbidity from this group. We are fortunate to have a large sample size in which these subgroups exist. We have further elaborated on this issue in our discussion by extending the following statement:

“This increasing frailty may have contributed to a reduced dialysis task participation during the course of the study among those who were undertaking more than 5 tasks at the outset, and baseline frailty could have prevented a subgroup of individuals with low task participation at baseline from increasing their task participation.”

I would like to see more detailed discussion on the putative reasons for not achieving primary outcome and if a different study design would have shown positive results.

We have added the following with references:

“Future interventions in this area should explore approaches to rehabilitate individuals whose self-efficacy has acutely or chronically declined, and intervention/trial designs that focus on incident patients and utilise Transitional Care Units to support individuals who are starting dialysis to learn tasks. (31)”

---

## [Editor Report · Decision Letter 1]

17 Jun 2021

A breakthrough series collaborative to increase patient participation with hemodialysis tasks: a stepped wedge cluster randomised controlled trial

PONE-D-21-06704R1

Dear Dr. Fotheringham,

We’re pleased to inform you that your manuscript has been judged scientifically suitable for publication and will be formally accepted for publication once it meets all outstanding technical requirements.

Kind regards,

Shahrad Taheri

Academic Editor

PLOS ONE
---

## [Editor Report · Acceptance letter]

9 Jul 2021

PONE-D-21-06704R1 

A breakthrough series collaborative to increase patient participation with hemodialysis tasks: A stepped wedge cluster randomised controlled trial 

Dear Dr. Fotheringham:

I'm pleased to inform you that your manuscript has been deemed suitable for publication in PLOS ONE. Congratulations! Your manuscript is now with our production department. 

Kind regards, 

on behalf of

Dr. Shahrad Taheri 

Academic Editor

PLOS ONE